# Mathematical Modeling of Clonal Interference by Density-Dependent Selection in Heterogeneous Cancer Cell Lines

**DOI:** 10.3390/cells12141849

**Published:** 2023-07-14

**Authors:** Thomas Veith, Andrew Schultz, Saeed Alahmari, Richard Beck, Joseph Johnson, Noemi Andor

**Affiliations:** 1Moffitt Cancer Center, Integrated Mathematical Oncology, USF Magnolia Drive, Tampa, FL 33612, USA; thomas.veith@moffitt.org (T.V.); andrew.schultz@moffitt.org (A.S.); richard.beck@moffitt.org (R.B.); 2Department of Cell Biology, Microbiology, and Molecular Biology, University of South Florida, 4202 E Fowler Ave, Tampa, FL 33612, USA; 3Department of Computer Science, Najran University, King Abdulaziz Road, Najran 61441, Saudi Arabia; saeed3@usf.edu; 4Moffitt Cancer Center, Analytic Microscopy Core, USF Magnolia Drive, Tampa, FL 33612, USA; joseph.johnson@moffitt.org

**Keywords:** mathematical oncology, tumor evolution, intratumoral heterogeneity, density-dependent selection, life history theory, r/K selection

## Abstract

Many cancer cell lines are aneuploid and heterogeneous, with multiple karyotypes co-existing within the same cell line. Karyotype heterogeneity has been shown to manifest phenotypically, thus affecting how cells respond to drugs or to minor differences in culture media. Knowing how to interpret karyotype heterogeneity phenotypically would give insights into cellular phenotypes before they unfold temporally. Here, we re-analyzed single cell RNA (scRNA) and scDNA sequencing data from eight stomach cancer cell lines by placing gene expression programs into a phenotypic context. Using live cell imaging, we quantified differences in the growth rate and contact inhibition between the eight cell lines and used these differences to prioritize the transcriptomic biomarkers of the growth rate and carrying capacity. Using these biomarkers, we found significant differences in the predicted growth rate or carrying capacity between multiple karyotypes detected within the same cell line. We used these predictions to simulate how the clonal composition of a cell line would change depending on density conditions during *in-vitro* experiments. Once validated, these models can aid in the design of experiments that steer evolution with density-dependent selection.

## 1. Introduction

Cellular heterogeneity is a defining feature of most cancers, and it is critical to tumor progression and treatment failure [1,2]. Advances in sequencing techniques have provided for an unprecedented depth of genetic profiling and ushered in a new era of individualized, data-driven cancer genetics [3]. Despite this progress, the relationship between genetic and phenotypic heterogeneity remains a significant gap in our current understanding.

One contributor to a cancer cell’s phenotype includes large-scale somatic copy number alterations (SCNAs) of 10 mega base pairs or more. Studies show that SCNAs correlate with progression rates and overall survival, with cancer cells grouped by copy number landscape exhibiting the same resistance to chemotherapy [4,5,6,7]. The SCNAs of whole chromosomes or chromosome arms, also known as aneuploidy, are a defining feature of many cancers [8]. Chromosomal instability (CIN) is a hallmark of cancer [9] and accounts for the vast majority of genetic material with an altered copy number state. CIN has been shown to promote metastasis and tumor evolution, particularly in cancers which have common aneuploidy patterns, such as gastric cancer [2,10]. Studies have shown that SCNAs activate oncogenes, disrupt tumor suppressor genes [11], correlate with cancer phenotype [12,13,14,15], and can be spatially segregated within the tumor [16]. Aneuploidy fuels rapid phenotypic evolution and drug resistance, with similar karyotype profiles displaying resistance to the same drug [17]. The correlation between karyotypic and phenotypic divergence is not surprising: two cells with different karyotypes will differ in the expression of thousands of genes, which in turn will have broad phenotypic effects. However, how selection acts upon karyotypes remains poorly understood.

Here, we aim to characterize a targeted subset of the phenotypic differences between co-existing karyotypes, which are defined by their ability to out-compete each other at low vs. high cell densities. Cell densities naturally vary in the tumor over space and time, thus creating niches with distinct selection pressures. A cell that is in a densely packed environment will be under more pressure to overcome contact inhibition than a cell that finds itself in sparse conditions. This variation in evolutionary pressures likely contributes to the coexistence of cancer cells with heterogeneous phenotypes. Density-dependent selection occurs when fitness is a function of population density [18]. A related concept in ecology is life history theory [19] and the r/K selection framework, which investigates trade-offs between the number of offspring a species produces (growth rate or ‘r’) and the ability to compete in dense ecological niches (carrying capacity or ‘K’) [20,21].

We present a framework for examining how density-dependent selection acts on coexisting clones defined by karyotypes and apply it to a set of stomach cancer cell lines. Hereby, we focus on carrying capacity as defined by spatial limitations rather than metabolic constraints.

## 2. Results

### 2.1. Identifying Biomarkers of Growth Rate and Carrying Capacity

By sequencing the DNA and RNA of >36,000 cells from nine stomach cancer cell lines, our prior work classified cells into groups with unique karyotypes [22], which are further referred to as superclones [23] or clones. To compare the growth dynamics across cell lines, we grew eight of these stomach cancer cell lines in a T25 flask until they reached confluence (6–23 days); we then imaged them every day to count cells (Section 4).

We changed the culture media every 3.05 days on average, with media changes becoming more frequent as cells became more confluent. We compared cell counts derived from a Countess (Life Technologies Countess II FL Automated Cell Counter) to cell counts derived from live cell imaging to confirm segmentation accuracy (Appendix A Figure A6), thus supporting feasibility of monitoring growth dynamics during routine *in-vitro* experiments.

We fit the Gompertz growth model and two instances of the generalized logistic growth model (Richards and Verhulst) [24] to this time-series data (Figure 1A; see also Section 4.4). The resulting $R^{2}$ was high (Adj-$R^{2}>0.95$) for each of the models and cell lines (Appendix A Table A1). Using the Akaike Information Criterion, we determined that the Richards model slightly outperformed the Verhulst model, which in turn slightly outperformed the Gompertz model (Appendix A Figure A3). We thus eliminated the Gompertz model from consideration for model selection. Using likelihood profiling [25] to assess practical identifiability, we concluded that the noise levels in the data collected for 4/8 cell lines were too high to infer all the parameters of the Richards model (Appendix A Figure A4). Therefore, we used the simpler Verhulst model (commonly referred to as the “logistic function”) to infer growth dynamics from time-series data for all eight cell lines (see Figure 1, Appendix A Table A2).

Overall, there was no significant correlation between the growth rate and the carrying capacity across cell lines (Pearson’s r=−0.37, p=0.54). In order to investigate the potential r/K trade-offs between clones within a cell line, we sought to find transcriptomic biomarkers of the inferred growth parameters in the scRNA-seq data available from our prior study [22]. Recently, investigators induced r/K-selection in HeLa cells [20]. The authors found that genes that were differentially expressed between r- and K-selected cells cultured at low densities were enriched among 25 pathways defined in the KEGG database [26]. We tested these pathways for their potential as biomarkers of the growth rate and carrying capacity in five of the eight gastric cancer cell lines (further referred to as the training set; Appendix A
Figure A2). For each of the 25 pathways, we fitted a linear regression model to predict the growth parameters inferred for a given cell from the cell’s respective pathway activity level. Because the cell cycle is one of the strongest modulators of pathway activity [22,27,28], we also grouped the cells according to their assigned cell cycle state, thereby calculating the median pathway activity across the following: (i) G0G1 cells, (ii) S cells, (iii) G2M cells, and (iv) all cell cycle states combined. The pathways that had the strongest predictive value for the growth rate (*r*) included the *‘Amoebiasis during G2M’* (adj-$R^{2}=0.89,p<0.02$), *‘Epstein–Barr virus infection’*, and *‘AMPK signaling pathway during S-phase’*. For the carrying capacity (*K*), the *‘PI3K-Akt signaling pathway’* (adj-$R^{2}=0.91,p<0.01$) and *‘Arginine and proline metabolism’* had the strongest signals (Appendix A
Figure A2). This process was repeated to include the remaining three cell lines (further referred to as the validation set), but only for these top performing pathways and cell cycle states (Figure 1B). Of these top pathways, 40% were confirmed in the validation set (FDR adjusted p<0.05; Figure 1B), including *‘Amoebiasis during G2M’* and *‘Arginine and proline metabolism’*, which were further used as proxies of the growth rate (*r*) and carrying capacity (*K*), respectively.

### 2.2. Towards Informing Future Experiments: Steering Clonal Evolution

The identification and characterization of co-existing clones within a cell line requires high-throughput assays, such as single-cell sequencing. Repeated measurements of a cell line’s clonal compositions at a high temporal resolution are thus cost-prohibitive. This underscores the need to identify biomarkers that can be used to estimate clonal growth parameters.

We previously identified 39 clones, which were each defined by distinct SCNAs and confirmed by both scDNA- and scRNA-seq data across the eight cell lines [22] (Figure 1E). Using pathway activity levels from the KEGG pathways *’Amoebiasis’* and in *’Arginine and proline metaboism’*, we predicted the growth rate and carrying capacity for every sequenced cell in each cell line. By grouping cells according to their clone membership (Figure 1F,G) we concluded that up to two r/K trade-offs may exist between clone pairs in two of the eight analyzed cell lines (Online Methods and Appendix A Table A3), namely in NCI-N87 and SNU-668 (Figure 2A).

While the growth conditions for SNU-668 can be modulated to favor the r-selected clone, by the time it exceeds the K-selected clone, its frequency in the cell line is less than 0.001%. In this case, the r/K trade-off between the two clones has little practical relevance, because other clones in the cell line take over the population faster than the two clones can outcompete each other (Figure 2B). In contrast, the r/K trade-off identified in NCI-N87 did have practical relevance, with most cell culture schedules resulting in populations where at least one of the two clones maintains high cell representation (>5%; Figure 2C).

Optimizing the seeding density and the timing for splitting cells should thus enable evolutionary steering of the cell line’s clonal composition across passages. A condition for such an optimal time to exist is that growth of the population can never be negative, which we prove analytically (Section A.2). We used the growth parameters predicted for the clones identified in the NCI-N87 cell line (Appendix A Table A3) to parameterize a multi-compartment ODE model to simulate how clonal composition changes over time (Figure 2B):(1)$$\frac{dN_{i}}{dt}=N_{i}*r_{i}\left(1-\frac{\sum_{i\in 1\ldots n}N_{i}}{K_{i}}\right),$$
where $N_{i}$ is the cell representation of clone *i*. $K_{i}$ and $r_{i}$ are the biomarker-inferred carrying capacity and maximum growth rate of each clone, respectively (see also Section 4.4).

The simulations shown in Figure 2D predict that shifts in clonal composition will likely occur as NCI-N87 cells grow *in-vitro*, with the magnitude of these shifts depending on the seeding density and the timing of splitting cells (Figure 2C–E). For example, with a low seeding density and a splitting interval of 7 days, the r-selected clone outcompeted the K-selected clone after only four passages (Figure 2E). By contrast, a high seeding density and a splitting interval of only 1 day maintained a relative dominance of the K-selected clone over >30 passages (Figure 2E). This suggests that, for a subset of cell lines, density conditions during *in-vitro* experiments can be optimized to either accelerate or delay changes in clonal composition.

## 3. Discussion

Our work rests on the shoulders of Uri Ben-David et al.’s landmark paper [29], which examined the responsiveness of 21 different variants of the same cell line with 321 anti-cancer agents. They found that 75% of the tested compounds that strongly inhibited some variants were inactive in others and that copy number changes (which are dominated by differences in karyotype) explained most of this differential phenotypic response. Two follow-up questions their work raises are: “*what is the mechanism of clonal interference for a specific culture condition?*” and “*can we predict the clonal composition that will emerge from a given culture condition?*”. Our work contributes to begin answering these multi-faceted questions. Through a re-analysis of existing scRNA-seq data from eight stomach cancer cell lines [22], we confirmed the biomarkers of growth rate and contact inhibition as previously identified in breast cancer cell lines [20], thus suggesting similarities in the mechanisms of contact inhibition exist across multiple tissue and cancer types. Differential expression of biomarkers for growth rate and contact inhibition/carrying capacity between clones within a cell line (Figure 1) suggest that different population densities will alter a cell line’s clonal composition. However, the presence of significant r/K trade-offs within cell lines was limited to only two clone pairs (Figure 2A). Most experimental protocols will not see the cells spend a long time at high density and will thus tend to select for cells with higher growth rates in the exponential phase, thus potentially explaining relative lack of significant r/K trade-offs within cell lines. Despite that, the predicted intra-cell line variability in both carrying capacity and maximum growth rate (Figure 1F,G) heralds ongoing changes in cellular composition, which are in line with changes in clonal evolution we observed previously over only 5–7 passages in a sub-set of these cell lines [22].

Our study focuses on a naturally fluctuating feature of the cellular microenvironment: population density. We therefore define phenotypic differences between karyotypes as their ability to out-compete each other at low vs. high population densities. However, we do not expect these traits to be the only phenotypic difference between the identified karyotypes. The expression magnitude of thousands of genes distinguishes any two karyotypes, and, with them, we expected multiple phenotypes to also differ. However, in the absence of any other differences in growth conditions (such as drug exposure), most phenotypic differences between karyotypes will be latent [29,30], thus allowing us to observe their relatively subtle differential growth under variable density conditions. We consider these differences to be only the tip of the iceberg among all phenotypic differences between co-existing karyotypes. However, the realization that even the most basic cell culture habits (seeding density and passaging frequency) can influence long-term cellular dynamics, we expect, will prompt the scientific community to prioritize developing databases that routinely record detailed cell culture protocols.

Clones that are far from each other in r–K space may also be spatially segregated and have differential treatment sensitivities. For example, the slower growth rate of K-selected cells may render them more resistant to cytotoxic therapies [31]—an effect that should be amplified by their spatial segregation [32,33]. We would expect K-selected sub-populations to be closer to the center of the tumor mass, where they are also more protected from drug penetration. By contrast, the enhancing tumor should be dominated by r-selected cells [31]. This scenario opens the door to an emerging concept of cancer treatment called adaptive therapy [34]. Adaptive regimens aim to optimize doses and dose schedules to maintain a fixed population of therapy-sensitive cells such that these, in turn, can suppress the growth of resistant cells [34]. Biomarkers that quantify the representation of r- and K-selected sub-populations could thus contribute to the design of adaptive regimens that spare a narrow border of r-selected cells such that the K-selected population stays enclosed [31]. The growing field of spatial transcriptomics makes it possible to measure the expression levels of thousands of genes throughout tissue space [35]. The ability to predict a cell’s placement along the r–K phenotype continuum from its transcriptome could thus facilitate the spatial de-lineation of r- from K-selected sub-populations in primary tumor tissue sections. Testing these ideas would require expanding our models into a spatial *in-vivo* framework. However, an understanding of the spatial distribution of r- and K-selected sub-populations could facilitate the design of therapy regimens, such as adaptive therapy or double bind therapy [36], aiming to stabilize the overall tumor burden to prevent or at least delay therapy resistance.

Our in silico results add to the evidence accumulating from many other studies [22,29,37] that every cell division is an opportunity for the cells to mutate and adapt to their environment. This insight underscores the need for routine tracking of the pedigree of evolving cell populations over decades, along with potentially changing environments (e.g., cell culture habits and therapy). Such efforts could help reveal long-term trends in the evolution of cell lines that remain elusive at shorter time-scales.

## 4. Online Methods

### 4.1. Cell Culture

The identity of each cell line was determined through independent karyotyping and mycoplasma contamination assessment. Cells were cultured in their recommended media conditions at 37°C. For, HGC-27, EMEM (Quality Biological Inc., Gaithersburg, MD, USA); KATOIII, RPMI-1640, and EMEM (1:1 mix) were used; for NCI-N87, RPMI-1640 (ATCC modified) was used; the remaining five cell lines (MKN-45, NUGC-4, SNU-601, SNU-638, and SNU-668) used RPMI-1640. All cell lines were grown in the aforementioned media with 10% fetal bovine serum (Gibco, Carlsbad, CA, USA) and 1% penicillin—streptomycin (Gibco). For cell passaging, Trypsin EDTA (0.25%) with phenol red (Gibco) was added, followed by inactivation using the respective growth media. Cells were seeded at a low initial density (range: 4.98–$38.64*10^{3}$ cells/cm^2^) in order to allow for the population’s exponential growth and subsequent plateau at the carrying capacity (range: 1.92–$15.32*10^{5}$ cells/cm^2^). The end-point for each experiment was determined by fitting a generalized logistic growth curve to the *in-vitro* data at various points during the time in culture. Across 8 cell lines, the time in culture ranged from 6–21 days, with media being refreshed on average every 48 h until the cells reached confluence, whereupon media was refreshed every 24 h. The experiment was stopped once the fold change in the inferred carrying capacity was less than 4% when removing any subset of the last 20% of time points.

### 4.2. Microscopy

Cells were seeded in a T25 flask (Fisherbrand), and sets of 4 phase contrast images were taken at 20× magnification for NCI-N87, and 10× magnification for the aforementioned cell lines on an Evos FL. Segmentation using Cellpose was utilized to quantify the cell count and cellular features (number of detections, centroid X μm, centroid Y μm, ROI, area m^2^, and perimeter μm). Growth curves were generated to examine the carrying capability of each cell line.

*Image preprocessing:* The microscopy image acquisition and light settings resulted in variations in the brightness of acquired images. The automatic segmentation and counting pipeline applied to dark images showed inaccurate results (i.e, mostly false negatives). Therefore, we applied pre-processing steps to our pipeline, which consisted of the following: (i) gamma correction to reduce darkness; (ii) histogram equalization to improve image contrast, and (iii) Gaussian blurring for smoothing.TV yes it should

*Cell segmentation:* Fully automated cell segmentation of phase contrast images was performed using Cellpose software.

Cellpose is a deep learning approach based on the U-Net architecture for cell segmentation [38,39], where vertical and horizontal spatial gradients of cells are predicted. Furthermore, Cellpose predicts a binary map of a cell location either inside or outside a region of interest (ROI). Using both the combined vertical and horizontal gradients and the binary map, the cell localization and generation of binary masks for every cell are performed. Our pipeline for cell segmentation using Cellpose used 2D microscopic images as input. An ROI annotation rectangle was applied to the phase contrast image (left corner index of the rectangle: (100,100); rectangle width and height: (1800 px, 1100 px). This ROI was used for subsequent image segmentation and analysis. We fine-tuned a pre-trained Cellpose model called (Cytotorch_2) for learning to segment cells on given microscopic images of different cell lines. After learning the segmentation using Cellpose, the trained model was tested on a hold-out set for evaluation. Then, feature extraction of each segmented cell was performed for further analysis of cell growth at a given time point. The features extracted from each cell segmentation included area, perimeter, roundness, and centroid. The pipeline saved the extracted features and visualization of the segmentation onto a user-desired folder for cell growth estimation and analysis.

### 4.3. Determination of Clonal Growth Parameters

Determining relative clonal carrying capacities, growth rates, and loss of contact inhibition parameter values is a necessary first step in modeling density-dependent selection in heterogeneous cell lines. To achieve this, we used gene expression signatures as surrogates of growth parameters as previously described [40,41].


*
Quantification of single cell pathway activity from gene expression:
*


In order to identify cells with active gene sets, we utilized AUCell version 3.18 for R version 4.3.1 [42]. AUCell takes as its input the scRNA sequencing data for the cells of interest and a list of gene sets. The output is the gene set activity for each cell. AUCell uses the area under the curve across the rankings of all genes of a particular cell, where genes are ranked by their expression value. A rank-based scoring method means AUCell is not affected by units or normalization methods of the gene expression data. Genes in the top 5% of the ranking are considered active in a given gene set. In order to account for potential batch effects, scRNA sequencing data for all cell lines were analyzed together. Cells with less than 200 features were excluded. Seurat version 2.3.4 was used to create a Seurat object for input to AUCell to quantify the activity of more than 2000 pathways from the KEGG database. For all further analysis, we focused on only a subset of 25 KEGG pathways, which were previously identified as being differentially expressed between r-selected and K-selected HeLa cells [20].


*
Clonal growth parameters:
*


We tested these 25 KEGG pathways quantified by AUCell as biomarkers of growth rate (*r*) and carrying capacity (*K*). The test takes the form of a linear model:(2)δ∼a∗p+c,
where δ is the parameter value (*r* or *K*), and *p* is the pathway activity. The pathways are then ranked by adjusted $R^{2}$, and the top five best-fitting pathways are prioritized as potential biomarkers of a given parameter.

With the linear models correlating pathway activity with growth parameters built at the cell line level, we can predict growth parameters for all clonal populations using their pathway activity as input (Figure 1E,F, Appendix A Table A3). Once parameters have been predicted, clonal growth can be simulated as systems of paired ODEs (see Section 4.4).

### 4.4. Mathematical Models of In-Vitro Cell Growth

The nature of tumor growth is not well known, and the exact laws which govern the growth of tumor cells will likely be context-dependent (cancer type, location in the body, etc.). However, even when many of these dependencies are held constant, it is difficult to discern between various models of cancer cell growth [43,44,45,46]. We fit generalized logistic growth models [24] to the time series cell growth data for eight gastric cancer cell lines (Appendix A Table A1):(3)$$\frac{dN_{i}}{dt}=N_{i}*r_{i}\left(1-{\left(\frac{\sum_{i\in 1\ldots n}N_{i}}{K_{i}}\right)}^{v}\right),$$
where *i* is either the entire cell line population (i.e., n=1) or one of multiple clones within a cell line (i.e., n≥3). $K_{i}$ and $r_{i}$ are the carrying capacity and maximum growth rate of each population, respectively. *v* is the loss of contact inhibition, which is assumed to be identical for all clones within a cell line. Two instances of Equation (Equation 3) were fitted to the data: one representing Richards growth model where *v* was kept variable and the other representing the logistic growth model with v:=1.

In addition to the two models mentioned above, we also fitted cell line population growth with the Gompertz model [24]. These models were chosen due to their prevalence in the literature for describing the growth of tumors and their biological interpretation. The growth data was read into R and fit using the package growthrates [47]. In order to compare the quality of growth-model fits, we calculated the adjusted-$R^{2}$ for each of the three models across all eight cell lines (Appendix A Table A1) We then compared Akaike Information Criterion scores (Appendix A Figure A3) and found the Richards model to have the lowest score in 7/8 cell lines. However, identifiability analysis revealed that we were unable to confidently infer the growth rate (r) and/or loss of contact inhibition (v) parameters in 4/8 cell lines (Appendix A Figure A4 and Figure A5). Thus, we modeled *in-vitro* growth using the logistic growth model. To model clonal growth dynamics, we used the ODE45 solver for MatLab version 9.7.0.1216025 to solve Equation (Equation 3) after being parameterized using the values in Appendix A Table A3.

### 4.5. Identifying r/K Trade-Offs between Co-Existing Clones within a Cell Line

Let $C=\{N_{1},\ldots N_{n}\}$ be the set of *n* clones identified within a given cell line. For each pair {x,y|x,y∈C∧x≠y} we use a Student’s *t*-test to compare the growth rates predicted for cells assigned to clone *x* ($r_{x}$) vs. cells assigned to clone *y* ($r_{y}$). We do the same for the carrying capacity predicted for cells of the two clones ($K_{x},K_{y}$). For clone pairs with a *p*-value ≤0.1 for both *r* and *K*, we further calculate $\tau_{r}\left(x,y\right)=\frac{\bar{r_{x}}}{\bar{r_{y}}}$ and $\tau_{k}\left(x,y\right)=\frac{\bar{K_{x}}}{\bar{K_{y}}}$. We define (x*,y*) as clones with potential r/K trade-offs:(4)$$\left(x*,y*\right)=\{x,y|\left(\tau_{r}\left(x,y\right)<1\wedge \tau_{k}\left(x,y\right)>1\right)\vee \left(\tau_{r}\left(x,y\right)>1\wedge \tau_{k}\left(x,y\right)<1\right)\}.$$

## Figures and Tables

**Figure 1 cells-12-01849-f001:**
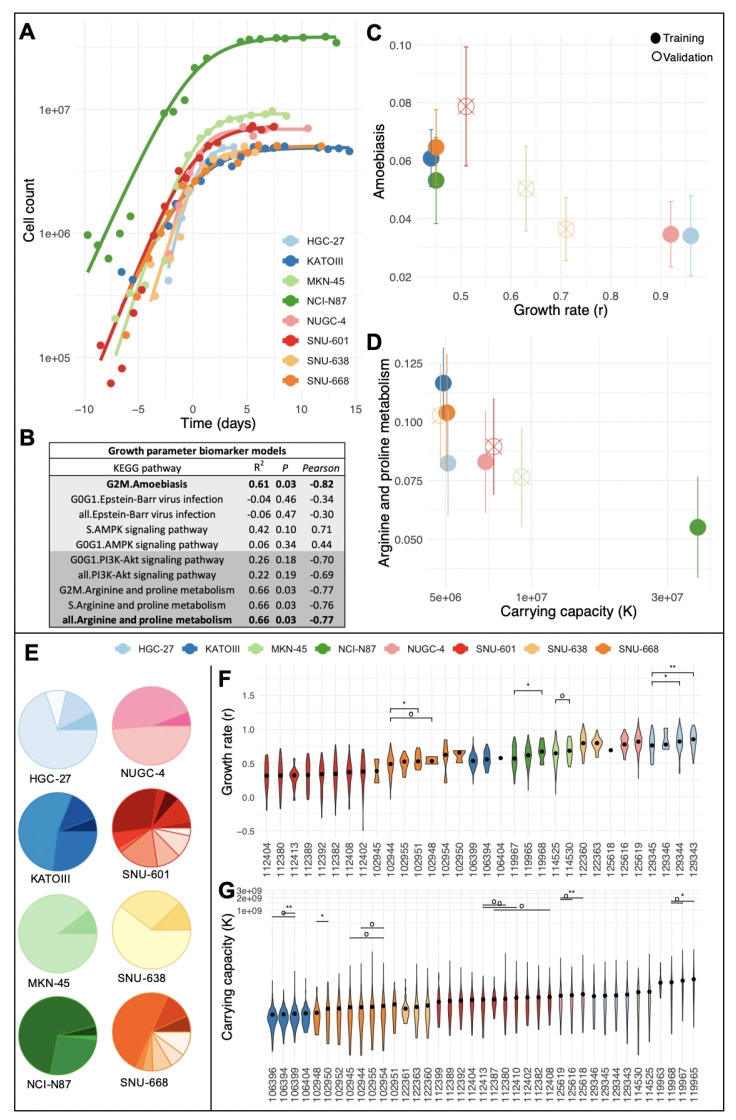
**Identifying biomarkers of growth and carrying capacity:** (**A**) Logistic growth curves fit to the cell counts of eight gastric cancer cell lines at various stages of their growth. Fits were shifted along the x-axis such that the midpoint of each curve lay above x=0. The eight cell lines differed in their maximum growth rate (*r*), as well as their maximum sustainable population size (*K*). (**B**) Summary statistics of linear regression models used to correlate KEGG pathway activity levels with logistic function growth parameters. The top five models and their performance in the training cell lines are shown for growth rate (*r*: light gray rows) and carrying capacity (*K*: dark gray). Columns: Pearson = Pearson correlation coefficient; P = FDR corrected *p*-value; $R^{2}$ = adjusted $R^{2}$. (**C**,**D**) Relation between growth parameters (r,K) and the model with best performance in the validation dataset: ‘Amoebiasis’ (**C**) and ‘Arginine and proline metabolism’ (**D**), respectively. Pathway activity quantified using AUCell with scRNA-seq data. Error bars represent median absolute deviation. (**E**) Clonal composition confirmed by both scDNA- and scRNA-seq in eight gastric cancer cell lines (data taken from [22]). (**F**,**G**) Violin plots showing predicted values for clonal growth rate as a function of ‘Amoebiasis’ activity (**F**), and clonal carrying capacity as a function of ‘Arginine and proline metabolism’ activity (**G**). Wilcoxon signed rank test: o *p* < 0.1, * *p* < 0.05, ** *p* < 0.01. If a cell line had more than three significant clone pairs, we displayed only the three highest *p*-values below 0.1.

**Figure 2 cells-12-01849-f002:**
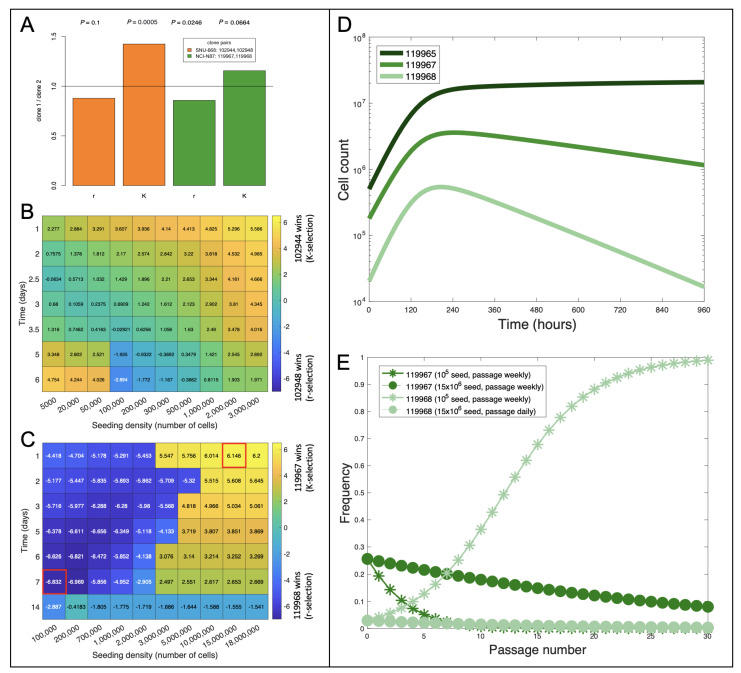
**Using biomarkers of growth and contact inhibition to steer clonal evolution across passages:** (**A**) Clone pairs with significant differences in both growth rates (*r*) and carrying capacities (*K*) were detected in two cell lines. Each color encodes for a single pair of clones (legend). Each bar is calculated as the ratio of *r* or *K* between the pair of clones shown in the legend. Horizontal line at 1 indicates identical parameters for both clones. Clone pairs with potential r/K trade-offs are represented by bars on both sides of the horizontal line. P-value of difference in *r* or *K* between a given pair of clones is indicated on the top of each bar. (**B**) Heat map showing outcome of clonal competition of the two clones shown in (**A**) for the SNU-668 cell line after 30 passages when grown at different seeding densities (x-axis) and passaging intervals (y-axis). Positive entries represent K selection ($Clone_{K}$ wins) and negative entries represent r selection ($Clone_{r}$ wins). The magnitude represents the size of the winning clone. Entries calculated as $sgn\left(Clone_{K}-Clone_{r}\right)*log_{10}\left(max\left(Clone_{r},Clone_{K}\right)\right)$. (**C**) Same as in (**B**), but for the NCI-N87 cell line. Note the presence of both large negative and large positive values for NCI-N87, in contrast to SNU-668, thus indicating the practical relevance of the r/K trade-off for the former. Highlighted in red are the conditions used for simulations in (**E**). (**D**) Simulated growth of the three largest clones (color-coded) detected in the NCI-N87 cell line over 40 days (smallest clone excluded due to insufficient G2M cell representation). (**E**) Change in frequency of the two NCI-N87 clones shown in (**A**) over multiple passages (x-axis). Changing the seeding density and the timing of splitting the cells between passages (stars vs. circles in legend), we predict will accelerate or delay the decline of clone 119967 (dark color shades), from passage 4 (intersection of star-shaped curves) to after passage 30 (circle-shaped curves). Clonal frequencies at harvest set the initial frequency conditions for each subsequent seeding.

## Data Availability

Supporting data for the reported results, as well as the code that generates the main figures in this manuscript, can be found at https://www.github.com/MathOnco/densityDependentSelection.

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
