# Peer review of "Mathematical Modeling of Clonal Interference by Density-Dependent Selection in Heterogeneous Cancer Cell Lines"

_cells, 2023, doi:10.3390/cells12141849_

Round 1

Reviewer 1 Report

Veith et al. apply mathematical modeling to previously published single cell RNA and DNA datasets from eight stomach cancer cell lines. They found transcriptomic biomarkers that indicated differential growth rates and carrying capacities among clones within cell lines. The authors then use this information to simulate clonal dynamics of one cell line, NCI-N87, comparing splitting times of one versus six days.

The authors’ goals to quantify cellular heterogeneity and clonal dynamics are both important pursuits, as is assessing the functional relevance of aneuploidy for cancer evolution. The acceleration or delay of clonal evolution is also very interesting and represents exciting opportunities for mathematical modeling and in vitro studies as the authors have shown here.

Below I list several additional comments and questions:

1.    Lines 46-47: How does defining the carrying capacity by spatial limitations rather than metabolic constraints impact these models? Can the authors be certain that the maximum sustainable population sizes in Figure 1A are not affected by metabolic constraints? This is also worth clarifying given that arginine and proline metabolism was found by the authors to have strong predictive value for carrying capacity.

2.    Line 103: The authors found putative r/K tradeoffs in two of the eight lines, but Figure 2B and 2C focus on one line (NCI-N87). Would the other line (SNU-668) undergo similar clonal dynamics?

3.    Lines 115-121: As the authors describe, these results show that the splitting time can impact competition of the two NCI-N87 clones. Rather than splitting time, would the splitting dilution (e.g. 1:2 versus 1:10) also have the same effect?

4.    Although I acknowledge the culturing would be a challenge, is it possible to validate the clonal dynamics in Figure 2C with an in vitro experiment of NCI-N87? Would the authors expect these two clones to have differential treatment sensitivities?

There are a few minor typos and unnecessary words in the manuscript.

Reviewer 2 Report

The brief report explores how density-dependent selection affects coexisting clones in stomach cancer cell lines. Through live-cell imaging, the authors identify variations in growth rates and contact inhibition, leading to the identification of important biomarkers related to growth rate and carrying capacity. The Verhulst model is used to analyze time-series data and infer growth dynamics. 

While the paper is well-written and almost everywhere clear, a few improvements could enhance its quality. Specifically, the lack of significant correlation between growth rate and carrying capacity across most cell lines and the limited presence of r/K tradeoff require further explanation. Can the multi-compartment model provide deeper insights into the tradeoff? Do shifts in clonal composition play a significant role? The paper should provide deeper commentary on these points. Additionally, discussing the study's perspectives and potential implications would be valuable.

Reviewer 3 Report

The main problem of this work is the actual absence of a significant result in it. The authors used a number of experimental and theoretical (simulation) methods for studying cell lines in vitro. The study is poorly targeted, and the result is best illustrated by a quote from the end of page three "This suggests that for a subset of cell lines the timing for splitting cells can be optimized to either accelerate or delay changes in clonal composition" The authors need to more clearly formulate the objectives and, most importantly, the positive results of the study, if any.

Round 2

Reviewer 1 Report

The authors addressed all of my previous questions and comments.

Author Response

Thank you.

Reviewer 2 Report

All the issues that I mentioned in my previous report have been addressed. I do not have any additional remarks. In my opinion, the paper is deserving of publication in Cells.

Author Response

Thank you.

Reviewer 3 Report

Even in a partially revised form, the article raises many questions about the formulation of the work and the significance and universality of the results obtained. Thus, the conclusion that within the same volume, the smaller the cell size, the greater their number (the K parameter of the logistic model) is obvious. Remarkably, it is confirmed by the results of scRNA-seq analysis.

In the very setting of the experiment with multiple transplantation of cells into fresh medium, the question of the advantage of clones with a high growth rate (r-selected subpopulations) over clones with a low growth rate but high density (K-selected subpopulations) is also quite obvious. The advantage of the latter could be manifested in conditions of limited resources, which corresponds to experiments with unfed culture conditions. So it is not surprising that the dominance of K-selected subpopulations is observed in only one case. If we assume that r- & K-selected subpopulations are characterized only by the corresponding correlations with pathways activities, and not with the values of the corresponding parameters of the logistic model, then the question arises about the significance of studying such clonal competition.

The discussion of the spatial heterogeneity of the cell population and its significance for therapy added by the authors is abstract, since the work itself does not contain experimental data on such heterogeneity, and the model under consideration is not spatially distributed and does not take into account cell motility.

In general, this article requires complete revision - it is necessary to throw out irrelevant information from it, such as the choice of a fitting mathematical model with all statistical parameters (based on them, all three models could be used). Realizing that the authors have done a lot of experimental work, it should be noted that its description does not provide anything for understanding the goals and results of the paper.

At the same time, there is no detailed description of the experimental design used, except for a reference to the work “Variation in the life history strategy underlies functional diversity of tumors. https://doi.org/10.1093/nsr/nwaa124” and the selection of relevant KEGG pathway biomarkers from Table 3 is not clear enough, so it raises more questions than answers.

The question of the setting/purpose of the experiment, as well as the results obtained and their significance, also remains open.

The article requires REAL major revision.
